# Differences in Tridimensional Shoulder Kinematics between Asymptomatic Subjects and Subjects Suffering from Rotator Cuff Tears by Means of Inertial Sensors: A Cross-Sectional Study

**DOI:** 10.3390/s23021012

**Published:** 2023-01-16

**Authors:** Cristina Roldán-Jiménez, Miguel Cuadros-Romero, Paul Bennett, Antonio I. Cuesta-Vargas

**Affiliations:** 1Department of Physiotherapy, Faculty of Health Sciences, Universidad de Malaga, 29016 Málaga, Spain; 2Instituto de Investigación Biomédica de Málaga (IBIMA), 29590 Málaga, Spain; 3Unit of Upper Limb Orthopedic Surgery of Hospital, University of Malaga, 29010 Málaga, Spain; 4School of Clinical Science, Faculty of Health Science, Queensland University Technology, Brisbane City, QLD 4059, Australia

**Keywords:** shoulder, assessment, kinematics, inertial sensors, motion analysis

## Abstract

Background: The aim of this study was to analyze differences in three-dimensional shoulder kinematics between asymptomatic subjects and patients who were diagnosed with rotator cuff tears. Methods: This cross-sectional study recruited 13 symptomatic subjects and 14 asymptomatic subjects. Data were obtained from three inertial sensors placed on the humerus, scapula and sternum. Kinematic data from the glenohumeral, scapulothoracic and thoracohumeral joints were also calculated. The participants performed shoulder abductions and flexions. The principal angles of movements and resultant vectors in each axis were studied. Results: The glenohumeral joint showed differences in abduction (*p* = 0.001) and flexion (*p* = 0.000), while differences in the scapulothoracic joint were only significant during flexion (*p* = 0.001). The asymptomatic group showed higher velocity values in all sensors for both movements, with the differences being significant (*p* < 0.007). Acceleration differences were found in the scapula during abduction (*p* = 0.001) and flexion (*p* = 0.014), as well as in the sternum only during shoulder abduction (*p* = 0.022). Conclusion: The results showed kinematic differences between the patients and asymptomatic subjects in terms of the mobility, velocity and acceleration variables, with lower values for the patients.

## 1. Introduction

The shoulder is exposed to degenerative physiological changes caused by aging, which affect motion-related structures such as the ligaments that stabilize the joint [1]. More specifically, degenerative changes can affect muscle structures, such as the rotator cuff [1]. The rotator cuff is formed by the supraspinatus, infraspinatus, minor round and subscapular muscles, constituting a functional unit that protects and stabilizes the joint at every point of its circumference [2]. The histological degeneration in the rotator cuff [3,4] that occurs over the years can lead to a tear or rupture due to the low vascularization [1].

In fact, it has been estimated that 36% of subjects with symptoms of the shoulder have a rotator cuff tear (RCT) [5], and the prevalence of RCT increases with age [6].

Hence, the upper limb’s range of motion (ROM) is a measure of interest in the clinical setting since it is essential for diagnosis, treatment evaluation and quantification of possible changes [7]. Besides ROM, the importance of focusing on kinematic aspects of shoulder movement, such as speed or acceleration [3,4] has been highlighted over the last few years. Nowadays, the three-dimensional (3D) study of shoulder kinematics is possible with inertial sensors, introduced by the aerospace industry as an accurate and reliable method for human mobility studies [4]. Regarding the shoulder, their reliability and validity have been reviewed, and several protocols have been developed for analyzing upper limb movements [8] and the scapulothoracic, thoracohumeral and elbow joints [9].

Recent research shows that shoulder evaluation in the clinical field is directed towards using devices such as inertial sensors [8]. These devices are accessible to all, practical and non-invasive, allowing a clinician to provide a measurement of motion quickly and easily that can be complemented by questionnaires such as disabilities of the arm, shoulder and hand (DASH) [10] or ULFI [11].

Although the description of shoulder movement characteristics includes angular displacement, speeds and accelerations [12], most research has focused on shoulder joint angles [13]. Furthermore, although shoulder kinematics has been widely studied, kinematics in shoulder lesions remains controversial, without a clear causal relationship between dyskinesia and a specific pathology [14].

The study aimed to compare the 3D shoulder kinematics of asymptomatic subjects and those waiting for RCT surgery, measured by inertial sensors. The hypothesis was that asymptomatic subjects have greater kinematic values.

## 2. Materials and Methods

### 2.1. Setting and Participants

This cross-sectional study recruited adult subjects suffering from RCT and adults without shoulder pain. Patients were recruited from a specialized orthopedics clinic where they had been previously diagnosed by magnetic resonance imaging. They were on a waiting list for RCT surgery. Asymptomatic subjects were recruited through advertisement. They were interested in participating in the project, and they met the inclusion criteria. Subjects were included if they were aged between 18 and 75 years old and had a body mass index (BMI) between 18 and 42. Patients were also excluded if their diagnosis had a different etiology from RCT. Asymptomatic subjects were excluded if they had shoulder pain or presented a positive Neer [15] or Hawkins [16] test. The subjects met all inclusion and exclusion criteria and gave informed consent to participate in the project. Ethical approval for the study was granted by the Ethics Committee of the Faculty of Health Sciences, University of Málaga. The principles of the Declaration of Helsinki were respected.

The RCT-affected group consisted of 13 subjects (male:female = 4:9, right-handed:left-handed = 11:2) with unilateral rotator cuff tears. Affected shoulders were measured: two right arms and eleven left arms. The asymptomatic group consisted of 14 subjects (male:female = 5:9, all of them right-handed). Twelve right arms and two left arms were measured, with both the right and left sides measured, like in the RCT-affected group.

The a priori sample size was calculated in nine patients for an α error of 0.05, statistical power of 0.8 and β error of 0.7, based on data from a systematic review of the use of inertial sensors to measure human movement [4].

### 2.2. Outcome Measures

Anthropometric and descriptive independent variables related to age, gender, weight, height and BMI were recorded. Nine physical properties were included, corresponding to three dependent variables: mobility angle (°), angular speed (°/s) and linear acceleration (m/s^2^) for each of three spatial axes: x, y and z.

Inertial measurements were obtained using three inertial sensors (InertiaCube3, Intersense Inc., Billerica, MA, USA) whose weight is 17 grams and dimensions are 26.2 mm × 39.2 mm × 14.8 mm. Each sensor contains an inertial three-degree of freedom orientation tracking system—yaw, pitch and roll (Euler angles RPY)—with an accuracy of 1 in yaw°, 0.25° in pitch and roll and angular range of 360°, able to detect an angular rate between 0° and 1200° per second, with a sampling frequency of 1000 Hz. The InterSense Server software by InertiaCube3™ provides a timestamp standard for all sensors, allowing for the data´s temporal alignment. The data were recorded by kinematic InterSense Server software, which was passed to a database using Microsoft^®^ Excel 2007.

### 2.3. Procedure

After recruitment, the participants attended the study at the Human Movement Laboratory, Faculty of Health Sciences (University of Málaga). Prior to attaching the sensors to the skin, inertial sensors were placed according to their body segment placement on a horizontal or vertical flat surface, so they were reset to zero using the InterSense Server software. Then, any body surfaces were cleaned with alcohol for better adhesion to the skin. Their placement was the pathological hemibody of each subject located on the middle third of the humerus, slightly posterior with the *Z* axis pointing away from the body, on the middle third of the upper spine of the scapula with the *X* axis aligned with the cranial edge of the scapular spine and on the flat part of the sternum, with the *Z* axis pointing away from the body (Figure 1) according to the protocol stablished by Cutti et al., whose validity and reliability have been stablished, having high accuracy and being highly correlated to the gold-standard optoelectronic system [9]. Fixation was ensured using double-sided adhesive tape attached to the patient’s skin to prevent slippage in flat areas. In cylindrical body segments, an adhesive bandage 5 cm wide (Strappal ^®^) was used (Figure 1).

Because of their positioning, the axes in each sensor correspond to different axes and planes of anatomical movement, shown in Appendix A.

Tasks were explained concisely and clearly so that the actions to perform were understood. The beginning and end of each task were indicated verbally by the researcher. Participants were told to perform the movements to the highest position they could reach without increasing their pain in RCT cases. They were asked to stand in body-neutral positions to perform the following kinematic tasks:Shoulder abduction (ABD) in the coronal plane, with the elbow extended, wrist in the neutral position, and the palm toward the midline at the beginning and end of the movement (four repetitions);Shoulder flexion (FLEX) in the sagittal plane, with the elbow extended, wrist in the neutral position, and the palm toward the midline at the beginning and end of the movement (four repetitions).

There was no pre-selected speed during the performance to study this variable.

To obtain information about shoulder disability in pathological subjects, the Spanish versions of the DASH [10] and ULFI [11] questionnaires were filled in by each participant before performing analytical tasks. The DASH questionnaire is a standardized measure of upper limb functional status and symptoms [17] consisting of a 30-item disability/symptom scale. It is valid and reliable for patients suffering from several upper limb disorders [18]. ULFI is an upper extremity outcome measure comprising a 25-item scale that can be transferred to a 100-point scale. It also has strong psychometric properties for reliability and validity [19]. The procedure is summarized in Figure 2.

### 2.4. Data Analysis

SPSS v22.0 was used for all statistical computations. Descriptive statistics (mean and standard deviation) were calculated for age, height, weight and BMI using standard procedures. An analysis of variance (one-way ANOVA with F and *p* values) and 95% CI were performed for the inertial variables to test for differences between the two groups. The Kolmogorov–Smirnov test showed a normal distribution of the data (*p* > 0.05). For all statistical comparisons, the α level was set at 0.05.

The second repetition of both analytical tasks was chosen for analysis. Besides obtaining angular mobility from each sensor, this variable was calculated for two different joints, representing motion between two body segments: glenohumeral, defined as the humerus relative to the scapula, and scapulothoracic, defined as the scapula relative to the sternum [20]. In the glenohumeral joint, scapula ME–LA degrees were rested on humerus AB–AD. In the case of the scapulothoracic joint, lateral sternum rotation was rested to scapular ME–LA for the ABD task, and sternum flexion–extension was rested to scapular AN–PO for the FLEX task.

Based on RAW data, the highest point the patient reached during arm elevation recorded in the sensor placed on the humerus was used as a cut-off point. The corresponding angles obtained at the lowest point reached by each sensor were subtracted to calculate angular mobility. No signal filter was applied to obtain the most significant amount of data.

For calculating velocity (°/s) and linear acceleration (m/s^2^), the norm of the resultant vector (Nrv) of the three axes of movement (*x, y, z*), calculated as *Nrv =*
$\sqrt{x^{2}+y^{2}+}z^{2}$, expressed velocity and acceleration variables.

## 3. Results

Subjects from both groups had similar ages, with a mean of 55.78 and 52.68 years in the healthy and pathologic groups, respectively. The pathologic group obtained a mean of 70.96 on the ULFI point scale and 63.14 in DASH, while the asymptomatic group obtained 0 points in both questionnaires. More details regarding descriptive and anthropometric variables are shown in Appendix B.

Regarding the joints’ angular mobility, there were highly significant differences between groups in the glenohumeral joint (*p* < 0.01 in both tasks). There were no significant differences in angular mobility of the scapulothoracic joint during ABD. However, there were significant differences during FLEX (*p* = 0.001). In both groups, these two joints showed greater mean mobility values during FLEX. More details are shown in Table 1.

Concerning angular mobility recorded independently by each sensor, highly significant differences (*p* < 0.01) were found between groups for both ABD and FLEX tasks. Asymptomatic subjects showed greater angular mobility in the humerus and scapula than those suffering from RCT. Those differences were significant in the humerus in the Y axis during both tasks. In the case of the scapula, the Z axis, representing ME-LA, was the only scapular movement with no significant differences. Mobility was also higher for the sternum in asymptomatic subjects, which also indicates more trunk compensation. Those differences were significant in all axes (Table 2).

When comparing velocity, greater values were found in the asymptomatic shoulder group, and the differences were highly significant in all sensors analyzed (*p* < 0.01). Regarding descriptive velocity data, in the patient group, velocity showed greater values during the ABD test in the scapula and sternum, while greater velocities were found during the FLEX exercise in the humerus. This differs from the asymptomatic group, in which velocity tended to be greater during the ABD (Table 3).

Regarding acceleration, there was no significant difference between groups in the humerus for either task. However, there were significant differences in acceleration in the scapula between groups for the ABD task (*p* = 0.001) and the FLEX task (*p* = 0.014), as well as in the sternum during the ABD task (see more details in Table 4).

## 4. Discussion

The present study analyzed differences in 3D shoulder kinematics between asymptomatic subjects and patients diagnosed with RCT in terms of mobility, velocity and acceleration, using three inertial sensors placed on the humerus, scapula and sternum. This allowed us to examine differences in mobility at the glenohumeral and scapulothoracic joints as well as in the scapula, humerus and sternum in terms of ROM but also velocity and acceleration.

Current research in the shoulder field is heading towards diagnoses based on kinematics due to some limitations of the generally accepted pathoanatomic model: on the one hand, a shoulder assessment using musculoskeletal tests aims to isolate the anatomical structure of interest that may be the cause of the problem. However, the shoulder constitutes a functional unit [21,22,23]. On the other hand, both the symptomatology and clinical expression of shoulder injuries vary widely [24,25,26]. Therefore, current research is focused on the characteristic movement impairments that are the cause or consequence of shoulder dysfunction, driving the development of a new movement system model [27]. Hence, although the results of the present study are broadly in line with those that support differences in kinematics between healthy and affected shoulders, the present research contributes to the establishment of good measurement standards for shoulder kinematics. More specifically, it reinforces the use of inertial sensors to analyze kinematics [9].

### 4.1. Body Segment´s Angular Mobility

Previous studies have examined shoulder kinematics using inertial devices. Jolles et al. [28] found that the mean humeral ROM was 123° for the ABD and 127.5° for the FLEX in a sample of pathological shoulders with a higher level of upper limb functionality (mean DASH score = 83.7; SD = 25.1). In our study, the humeral angles were 77.55° and 88.48° for ABD and FLEX, respectively. The smaller observed ROM may have been because the pathological sample of Jolles et al. was composed of subjects suffering from injuries with etiologies other than RTC and because patients in the present study had a lower level of upper limb functionality (mean DASH score = 60.89; SD = 24.54). Duc et al. [29] measured kinematic variables with two inertial sensors placed on the humerus and sternum during daily living in 41 healthy subjects and 21 suffering from RCT (before and after surgery) to measure functional activity under everyday conditions in healthy and pathological shoulders. Körver et al. [30] could differentiate with high diagnostic power between healthy subjects and those suffering from shoulder pathologies (mean DASH score = 51; SD = 20) in functional activities using one inertial sensor placed on the humerus. That study established the use of inertial sensors to study shoulder kinematics in asymptomatic and pathological shoulders, although the results are not comparable with those of our study because of differences in data collection in both studies.

### 4.2. Scapulothoracic and Glenohumeral Joints’ Angular Mobility

Electromagnetic sensors have also been used to study kinematic differences between asymptomatic and symptomatic shoulders (mean DASH score = 21.4; SD = 10.8). Regarding the scapulothoracic joint [31], symptomatic subjects were found to have less upward scapular rotation in arm elevation. However, there were no significant differences during FLEX. This is in contrast with our study, in which no differences were found during ABD. In that study, scapular ME-LA showed a high variability between participants. This is in line with our study, as the Z axis, representing scapular ME-LA, was the only axis in which there were no differences between groups. Regarding the glenohumeral joint [32], there were differences in ABD and FLEX, which varied depending on the amount of elevation performed, confirming the kinematic differences between asymptomatic and symptomatic shoulders. This coincides with glenohumeral motion in our study, in which greater ROM was found in the asymptomatic group (*p* < 0.001) for both movements, despite differences in the DASH scores of symptomatic subjects between studies. Three-dimensional scapular kinematics have also been studied using Polhemus Fastrak in healthy subjects and patients with osteoarthritis and frozen shoulder (mean DASH score = 51.8; SD = 13.9) [33], showing increased scapular lateral rotation as a compensatory mechanism in pathological shoulders. This is in contrast with the present study, in which patients showed a greater value of ME-LA than asymptomatic subjects only during FLEX, although the differences were not significant (*p* = 0.59). Another study using the Polhemus Fastrak device with a similar symptomatic sample (mean DASH score = 70.1; SD = 19.7) [34] found a greater arm elevation amplitude in the ABD (mean =128.8; SD = 29.9) than in the FLEX (mean = 112.3; SD = 35.8) in patients suffering from RCT, in contrast with the results of Jolles et al. [28] and the present study.

### 4.3. Acceleration

To the best of our knowledge, this is the first study to report scapula acceleration. Only a few studies have studied kinematic scores based on other variables, including acceleration and clinical questionnaires [28,35]. Acceleration data allow for a more specialized comparative analysis between populations. For example, differences in mobility and velocity are found between RCT patients and asymptomatic patients in both the humerus and the scapula. However, focusing on acceleration, differences can only be seen in the scapula. These results highlight the importance of this variable when the shoulders are affected.

### 4.4. Clinical Implications and Future Perspectives

Previous studies have shown that humerus and scapular kinematics can explain variability in upper limb function [36]. As patients from the present study were on a waiting list for RCT surgery, the present measures could be used to give priority to subjects who present a greater alteration in the kinematics and function of the shoulder. On the other hand, while this study focuses on kinematic data captured by an inertial sensor, these instruments can be used as a complement to other devices, such as RGB-D cameras to achieve higher accuracy [37], optimizing their advantages [38]. In addition, data from inertial sensors can be used to develop a hybrid deep learning (HDL) system to facilitate the rehabilitation process [39].

### 4.5. Limitations

This study presents some limitations: it should be considered that including subjects of similar ages in the control (mean = 55.78; SD = 9.45 years) and RCT groups (mean = 56.69; SD = 7.76 years) may have resulted in the inclusion of asymptomatic subjects who were, in fact, suffering from RCT, as RCT abnormalities become more prevalent with age [6].

Another limitation is that the study focused on analytical tasks but did not analyze any functional task common in other studies, such as hair-combing (hand–head) or washing the back (hand–back). Finally, this study employed inertial sensors. These apparatuses are non-invasive, reliable and accurate tools used to analyze motion [4], but one of their limitations is that placing sensors over the skin can create soft-tissue artifacts [8].

Although both shoulder kinematics and the factors involved in RCT and the presence of symptoms remain unknown, these limitations should be addressed in future research, for example, by including an asymptomatic group with imaging tests or by comparing kinematic differences between healthy and affected shoulders classified by age.

## 5. Conclusions

This study analyzed shoulder kinematic differences between asymptomatic subjects and patients diagnosed with RCT who were waiting for surgery. This allowed us to examine objective variables that qualify motion in addition to quantifying it, such as velocity and acceleration. The kinematics results, complemented with subjective self-reported disability provided by the questionnaires, may help clinicians prioritize the order of waiting lists according to the degree of severity, heading towards a new movement system model in shoulder assessment. Future research should include the analysis of functional tasks as well as a wider sample that would allow for classification by age.

## Figures and Tables

**Figure 1 sensors-23-01012-f001:**
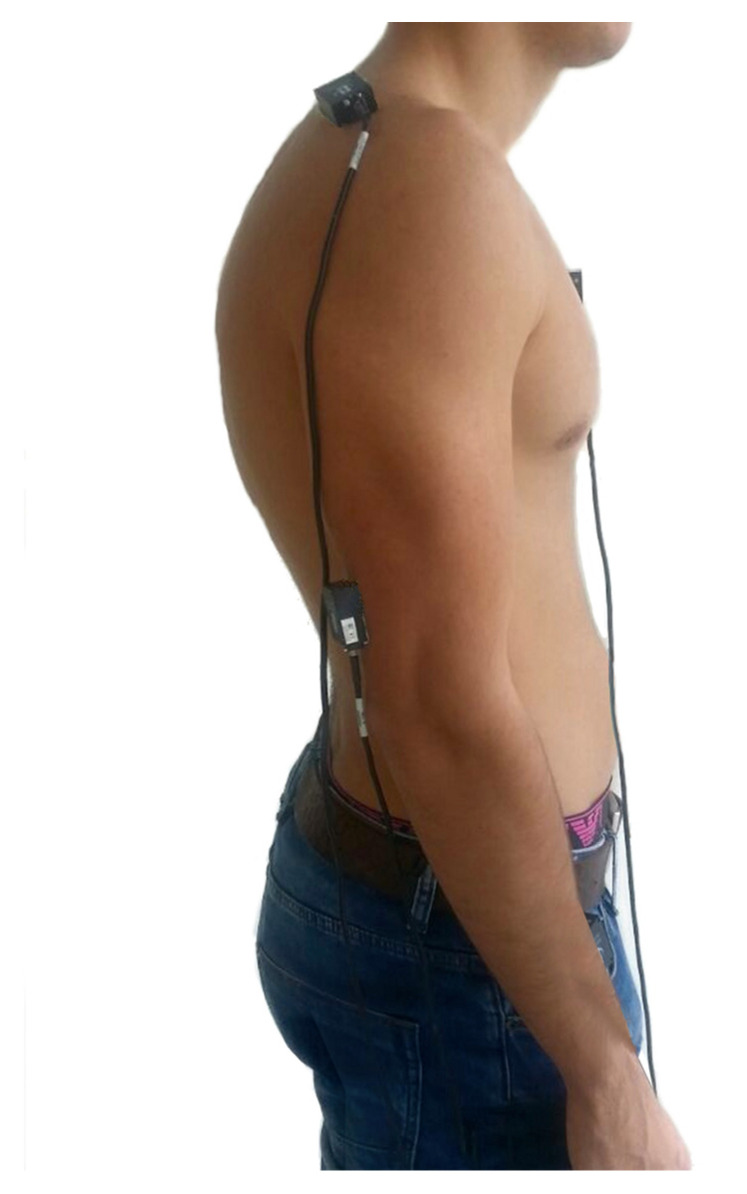
Inertial sensors’ placement.

**Figure 2 sensors-23-01012-f002:**
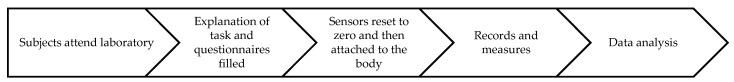
The procedure of the study.

**Table 1 sensors-23-01012-t001:** Mean (95% CI) joint angular mobility (°) from each axis in abduction and flexion.

Mobility	Glenohumeral (Mean, 95% CI)			Scapulothoracic (Mean, 95% CI)		
Group	Asymptomatics	Patients	ANOVA (F, *p*)	Asymptomatics	Patients	ANOVA (F, *p*)
ABD	133.35 (124.60–142.11)	72.97 (44.65–101.30)	15.614; 0.001	−8.76 (−17.10–−0.42)	1.12 (−5.95–−8.19)	4.415; 0.56
FLEX	137.69 (128.91–146.47)	82.10 (54.58–109.62)	19.718; <0.001	8.01 (12.91–22.60)	6 (0.80−11.21)	13.212; 0.001

**Table 2 sensors-23-01012-t002:** Mean (95% CI) sensor angular mobility (°) from each axis in abduction and flexion movement.

Axis	Humerus			Scapula			Sternum		
	Asymptomatics	Patients	ANOVA (F, P)	Asymptomatics	Patients	ANOVA (F, P)	Asymptomatics	Patients	ANOVA (F, P)
ABD X	65.48 (34.12–96.85)	38.01 (16.17–59.85)	2.452; 0.13	65.48 (34.12–96.85)	38.01 (16.17–59.85)	4.686; 0.04	14.65 (11.33–17.91)	5.71 (1.69–9.73)	14.280; 0.001
ABD Y	128.60 (108.18–149.02)	77.557 (48.18–106.92)	9.668; 0.005	33.83 (27.52–40.14)	12.63 (5.56–19.70)	22.764; <0.001	4.48 (2.45–6.51)	1.99 (0.37–3.61)	4.457; 0.05
ABD Z	54.52 (31.14–77.89)	35.72 (21.12–50.31)	2.209; 0.15	5.73 (2.52–8.95)	4.57 (1.87–7.28)	0.377; 0.54	14.58 (10.05–19.64)	3.74 (−2.77–10.26)	9.116; 0.006
FLEX X	86.45 (45.45–127.45)	60.88 (17.42–104.34)	0.883; 0.36	23.25 (18.75–27.75)	8.30 (1–15.60)	15.883; 0.001	12.29 (8.57–16)	3.74 (0.65–6.8)	15.883; 0.001
FLEX Y	142.74 (134.66–150.82)	88.48 (60.75–116.21)	20.112; <0.001	29.09 (25.05–33.13)	11.53 (5.15–17.91)	28.126, <0.001	5.31 (3.42–7.20)	2.29 (0.01–4.56)	5.100; 0.03
FLEX Z	58.92 (37.31–80.54)	40.93 (25.61–56.24)	2.066, 0.16	5.4 (3.51–7.40)	6.38 (2.98–9.78)	0.293; 0.59	12.78 (8.07–17.49)	3.86 (−0.66–8.40)	8.527; 0.008

**Table 3 sensors-23-01012-t003:** Mean (95% CI) velocity (°/s) from the resultant vector in abduction and flexion movement.

Axis	Humerus			Scapula			Sternum		
	Asymptomatics	Patients	ANOVA (F, P)	Asymptomatics	Patients	ANOVA (F, P)	Asymptomatics	Patients	ANOVA (F, P)
ABD Nrv	226.63 (190.26–263)	109.42 (70.80–147.98)	23.051; <0.001	56.85 (45.18–68.52)	43.71 (27.97–59.45)	17.875; <0.001	32.15 (28.63–35.66)	22.56 (15.96–29.17)	8.528; 0.007
FLEX Nrv	242.10 (205.67–278.52)	110.23 (70.54–149.93)	29.030; <0.001	92.19 (83.22–101.16)	40.99 (24.25–57.73)	38.813; <0.001	39.03 (32.90–45.17)	22.39 (14.73–30.34)	14.196; 0.001

**Table 4 sensors-23-01012-t004:** Mean (95% CI) acceleration (m/s^2^) from the resultant vector in abduction and flexion movement.

Axis	Humerus			Scapula			Sternum		
	Asymptomatics	Patients	ANOVA (F, P)	Asymptomatics	Patients	ANOVA (F, P)	Asymptomatics	Patients	ANOVA (F, P)
ABD Nrv	22.30 (21.14–23.45)	16.25 (9.86–222.64)	4.119; 0.05	8.27 (7.51–9.03)	4.48 (2.72–6.24)	16.468; 0.001	3.56 (2.05–5.07)	1.64 (1.15–2.13)	6.001; 0.02
FLEX Nrv	22.14 (21.48–22.8)	18.53 (8.92–28.14)	0.828; 0.37	8.14 (7.51–8.78)	5.01 (2.25–7.77)	7.065; 0.014	3.26 (2.19–4.32)	2.38 (0.99–3.76)	1.268; 0.27

## Data Availability

Data will be available upon reasonable request to the corresponding author.

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
