# Peer review of "Differences in Tridimensional Shoulder Kinematics between Asymptomatic Subjects and Subjects Suffering from Rotator Cuff Tears by Means of Inertial Sensors: A Cross-Sectional Study"

_sensors, 2023, doi:10.3390/s23021012_

Round 1

Reviewer 1 Report

Overall, interesting study entitled Differences in Tridimensional Shoulder Kinematics Between Asymptomatic Subjects and Subjects Suffering From Rotator Cuff Tears By Means of Inertial Sensors: a Cross-Sectional Study.

 Comments to authors:

·         Please make sure that the structure for citing published literature in the text, as well as the style of references in the References section, are consistent with the journal's style (see Instructions to Authors).

·         English language needs revision for style and syntax.

·         Abstract must be rewritten. Add characteristics of the participants (age…..) I suggest focusing the abstract on your study and your results.

·         Please add the originality of the study and add hypothesis at the end of the introduction section. Be please be more specific

·         Include more characteristics of participants. More information about the participant’s selection needed.

·         Please specify inclusion/exclusion criteria. The experimental protocol is not clear. Add chart flow.

·         Please justify the sample size.

·         Discussion: describing each part of the study

·         Please discuss the results of the study in relation to the previous studies.

Author Response

RESPONSE TO REVIEWER 1:

Comments and Suggestions for Authors

Overall, interesting study entitled Differences in Tridimensional Shoulder Kinematics Between Asymptomatic Subjects and Subjects Suffering From Rotator Cuff Tears By Means of Inertial Sensors: a Cross-Sectional Study.

 Authors: Thank you.

 Comments to authors:

  • Please make sure that the structure for citing published literature in the text, as well as the style of references in the References section, are consistent with the journal's style (see Instructions to Authors).

      Authors: Thank you. References are inserted with “Zotero”, and its preference is set to Sensors Journal.

  • English language needs revision for style and syntax. 

Authors: Thank you. The whole manuscript has been proof read and revised. All changes can be seen in the tracked version of the manuscript.

  • Abstract must be rewritten. Add characteristics of the participants (age…..) I suggest focusing the abstract on your study and your results.

Authors: Thank you. The abstract has been modified. Now, it can be read was follows:

Background: The aim of this study was to analyze differences in three-dimensional shoulder kinematics between asymptomatic subjects and patients who were diagnosed with rotator cuff tears. Methods: This cross-sectional study recruited 13 symptomatic subjects and 14 asymptomatic subjects. Data were obtained from three inertial sensors placed on the humerus, scapula and sternum. Kinematic data from glenohumeral, scapulothoracic and thoracohumeral joints were also calculated. Participants performed shoulder abduction and flexion. Principal angles of movements and resultant vectors in each axis were studied Results: Glenohumeral joint showed differences in abduction (P= .001) and flexion (P = .000), while differences in scapulothoracic joint were only significant during flexion (P = .001). Asymptomatic group showed higher velocity values in all sensors for both movements, being differences significant (P > .007). Acceleration differences were found in scapula in abduction (P = .001) and flexion (P = .014), as well as in sternum only during shoulder abduction (P = .022). Conclusion: Results showed kinematic differences between patients and asymptomatic subjects in mobility, velocity and acceleration variables, being values lower for patients.”

  • Please add the originality of the study and add hypothesis at the end of the introduction section. Be please be more specific

Authors: Thank you. The last parapgrah of introduction has been rewritten as follows:

The study aimed to compare 3D shoulder kinematics measured by inertial sensors between asymptomatic subjects and waiting for RCT surgery. The hypothesis was that asymptomatic subjects have greater kinematic values.”

  • Include more characteristics of participants. More information about the participant’s selection needed.

Authors: Thank you. The information was already provided in the manuscript, but it was wrongly placed. Now, the first pharagraph from “setting and participants” can be read as follows:

This cross-sectional study recruited adult subjects suffering from RCT and adults without shoulder pain. Patients were recruited from a specialized orthopaedics clinic where they had been previously diagnosed by magnetic resonance imaging. They were on a waiting list for RCT surgery. Asymptomatic subjects were recruited through advertisement. They were interested in participating in the project and they met the inclusion criteria .Subjects were included if they were aged between 18 and 75 years old, and had a Body Mass Index (BMI) between 18 and 42. Patients were also excluded if their diagnosis had a different aetiology from RCT. Asymptomatic subjects were excluded if they had shoulder pain or presented a positive Neer [15] or Hawkins [16] tests. The subjects met all inclusion and exclusion criteria and gave informed consent to participate in the project”

  • Please specify inclusion/exclusion criteria. The experimental protocol is not clear. Add chart flow.

Authors: Thank you. Inclusion and exclusion criteria is provided in the previous comment. Regarding the protocol, it has meed modified: We have added new details about inertial sensors, the measures and the procotocol. All of them can be seen in tracked changes in the document. For example:

Data was recorded by kinematic Intersense Server Software, which was passed to a database of Microsoft® Excel 2007”

“Prior to attaching sensors to the skin, inertial sensors were placed, according to their body segment placement, on a horizontal or vertical flat surface, so they were reset at 0 using the Intersense Server Software”

“Fixation was ensured using a double-sided adhesive tape attached to the patient's skin to prevent slippage in flat areas. In cylindrical body segments, an adhesive bandage 5 cm wide (Strappal ®) was used (figure 1)”

Tasks were explained concisely and clearly, so the actions to perform were understood. The beginning and end of each task were indicated verbally by the researcher.”

They were placed standing in body neutral position to perform the following kinematic tasks:

  1. Shoulder abduction (ABD) in the coronal plane, with the elbow extended, wrist in the neutral position, and the palm toward the midline at the beginning and end of the movement (four repetitions).
  2. Shoulder flexion (FLEX) in the sagittal plane, with the elbow extended, wrist in the neutral position, and the palm toward the midline at the beginning and end of the movement (four repetitions)”

As you suggested, we added a new figure 2:

Figure 2. The procedure of the study.

  • Please justify the sample size.

Authors: Thank you. The sample size was already provided in the manuscript:

Priori sample size was calculated in 9 patients for an α error of 0.05, statistical power of 0.8 and β error of 0.7, based on data from a systematic review on the use of inertial sensors to measure human movement [6].”

  • Discussion: describing each part of the study

Authors: Thank you. Following your suggestion, we have added the following subheadings:

Body segment´s angular mobility.

Scapulothoracic and glenohumeral joint´s angular mobility.

Acceleration.

Clinical implications and future perspectives.

Limitations

  • Please discuss the results of the study in relation to the previous studies.

Authors: Thank you. We have added the new paragraph:

Clinical implications and future perspectives.

Previous studies have shown that humerus and scapular kinematics can explain variability in upper limbs function [37]. As patients from the present study were in waiting list for RCT surgery, the present measures could be used to determine to give priority to those subjects who present a greater alteration of the kinematics and function of the shoulder. On the other hand, while this study focuses on kinematic data captured by an inertial sensor, these instruments can be used as a complement to other devices, such as RGB-D cameras to achieve higher accuracy [38] and optimizing their advantages [39]. In addition, data from inertial sensors can be used to develop hybrid deep learning (HDL) system to facilitate rehabilitation process [40].

New references:

  1. Roldán-Jiménez, C.; Cuadros-Romero, M.; Bennett, P.; McPhail, S.; Kerr, G.K.; Cuesta-Vargas, A.I.; Martin-Martin, J. Assessment of Abduction Motion in Patients with Rotator Cuff Tears: An Analysis Based on Inertial Sensors. BMC Musculoskelet. Disord. 2019, 20, 597, doi:10.1186/s12891-019-2987-0.
  2. Zhao, X.; Miao, C.; Zhang, H. Multi-Feature Nonlinear Optimization Motion Estimation Based on RGB-D and Inertial Fusion. Sensors 2020, 20, 4666, doi:10.3390/s20174666.
  3. Gosala, N.B.; Wang, F.; Cui, Z.; Liang, H.; Glauser, O.; Wu, S.; Sorkine-Hornung, O. Self-Calibrated Multi-Sensor Wearable for Hand Tracking and Modeling. IEEE Trans. Vis. Comput. Graph. 2021, PP, doi:10.1109/TVCG.2021.3131230.
  4. Bijalwan, V.; Semwal, V.B.; Singh, G.; Mandal, T.K. HDL-PSR: Modelling Spatio-Temporal Features Using Hybrid Deep Learning Approach for Post-Stroke Rehabilitation. Neural Process. Lett. 2022, doi:10.1007/s11063-022-10744-6.

Reviewer 2 Report

1. Please declare that You have made Ethical Committee for the study and send their report on dataset collection and experimental procedure, and declare that no humans or animals were harmed during the dataset collection and experiment.

2. Why did you use an Inertial measurement sensor? Its data is too noisy. For motion analysis, you may use the RGBD camera like Microsoft Azure or ZED 2i. It has precise and great accuracy.

3. Kindly highlight the procedure and how did you deal with the data, like data preprocessing and removing nan values. The procedure is not clearly explained. Rewrite the procedure again. Add some recent literature reviews of related works.

4. Align the Abstract, Introduction and conclusion.

5. Enhance the quality of the english writing of the paper and make it easily understandable and better for readers.

5. Read this article; this may help you to find a new innovative method as per your requirement.

Bijalwan, V., Semwal, V. B., Singh, G., & Mandal, T. K. (2022). HDL-PSR: Modelling Spatio-Temporal Features Using Hybrid Deep Learning Approach for Post-Stroke Rehabilitation. Neural Processing Letters, 1-20.

Author Response

RESPONSE TO REVIEWER 2:

Comments and Suggestions for Authors

  1. Please declare that You have made Ethical Committee for the study and send their report on dataset collection and experimental procedure, and declare that no humans or animals were harmed during the dataset collection and experiment.

Authors: Thank you. This is already done. Please see the following paragraph:

Settings and participants section:

“The subjects met all inclusion and exclusion criteria and gave informed consent to participate in the project. Ethical approval for the study was granted by the Ethics Committee of the Faculty of Health Sciences, University of Málaga. The principles of the Declaration of Helsinki were respected.”

At the end of the manuscritpt:

“Institutional Review Board Statement: The study was conducted in accordance with the Declaration of Helsinki, and approved by Ethics Committee of the Faculty of Health Sciences, University of Málaga “

  1. Why did you use an Inertial measurement sensor? Its data is too noisy. For motion analysis, you may use the RGBD camera like Microsoft Azure or ZED 2i. It has precise and great accuracy.

Authors: Thank you. All instruments have advantages and disadvantages. Inertial sensors have shown to be valid and reliable. Both RGB-D camera and intertial sensors are precise and accuracy. In fact, they are not opposite and they can complement each other. This information has been included in the discussion section, as follows:

“…On the other hand, while this study focuses on kinematic data captured by an inertial sensor, these instruments can be used as a complement to other devices, such as RGB-D cameras to achieve higher accuracy [38] and optimizing their advantages [39]

New references:

  1. Zhao, X.; Miao, C.; Zhang, H. Multi-Feature Nonlinear Optimization Motion Estimation Based on RGB-D and Inertial Fusion. Sensors 2020, 20, 4666, doi:10.3390/s20174666.
  2. Gosala, N.B.; Wang, F.; Cui, Z.; Liang, H.; Glauser, O.; Wu, S.; Sorkine-Hornung, O. Self-Calibrated Multi-Sensor Wearable for Hand Tracking and Modeling. IEEE Trans. Vis. Comput. Graph. 2021, PP, doi:10.1109/TVCG.2021.3131230.

  1. Kindly highlight the procedure and how did you deal with the data, like data preprocessing and removing nan values. The procedure is not clearly explained. Rewrite the procedure again. Add some recent literature reviews of related works.

Authors: Thank you. The procedure has been rewritten and reorderer. We have add new details, for example:

Data was recorded by kinematic Intersense Server Software, which was passed to a database of Microsoft® Excel 2007”

“Prior to attaching sensors to the skin, inertial sensors were placed, according to their body segment placement, on a horizontal or vertical flat surface, so they were reset at 0 using the Intersense Server Software”

“Fixation was ensured using a double-sided adhesive tape attached to the patient's skin to prevent slippage in flat areas. In cylindrical body segments, an adhesive bandage 5 cm wide (Strappal ®) was used (figure 1)”

Tasks were explained concisely and clearly, so the actions to perform were understood. The beginning and end of each task were indicated verbally by the researcher.”

They were placed standing in body neutral position to perform the following kinematic tasks:

  1. Shoulder abduction (ABD) in the coronal plane, with the elbow extended, wrist in the neutral position, and the palm toward the midline at the beginning and end of the movement (four repetitions).
  2. Shoulder flexion (FLEX) in the sagittal plane, with the elbow extended, wrist in the neutral position, and the palm toward the midline at the beginning and end of the movement (four repetitions)”

Also, in data analysis:

“Based on RAW data, the highest point the patient reached during arm elevation recorded in the sensor placed on the humerus was used as a cut-off point. The corre-sponding angles obtained at the lowest point reached by each sensor were subtracted to calculate angular mobility. No signal filter was applied to obtain the most significant number of data”.

  1. Align the Abstract, Introduction and conclusion.

Authors: Thank you. The abstract has been modified. Now, it can be read was follows:

Background: The aim of this study was to analyze differences in three-dimensional shoulder kinematics between asymptomatic subjects and patients who were diagnosed with rotator cuff tears. Methods: This cross-sectional study recruited 13 symptomatic subjects and 14 asymptomatic subjects. Data were obtained from three inertial sensors placed on the humerus, scapula and sternum. Kinematic data from glenohumeral, scapulothoracic and thoracohumeral joints were also calculated. Participants performed shoulder abduction and flexion. Principal angles of movements and resultant vectors in each axis were studied Results: Glenohumeral joint showed differences in abduction (P= .001) and flexion (P = .000), while differences in scapulothoracic joint were only significant during flexion (P = .001). Asymptomatic group showed higher velocity values in all sensors for both movements, being differences significant (P > .007). Acceleration differences were found in scapula in abduction (P = .001) and flexion (P = .014), as well as in sternum only during shoulder abduction (P = .022). Conclusion: Results showed kinematic differences between patients and asymptomatic subjects in mobility, velocity and acceleration variables, being values lower for patients.”

  1. Enhance the quality of the english writing of the paper and make it easily understandable and better for readers.

Authors: Thank you so much. This comment is closely related to a comment made by reviewer 1. The whole manuscript has been proof read and revised. All changes can be seen in the tracked version of the manuscript.

Read this article; this may help you to find a new innovative method as per your requirement.

Bijalwan, V., Semwal, V. B., Singh, G., & Mandal, T. K. (2022). HDL-PSR: Modelling Spatio-Temporal Features Using Hybrid Deep Learning Approach for Post-Stroke Rehabilitation. Neural Processing Letters, 1-20.

Authors: Thank you. It has been included in the manuscript, as folloes:

In addition, data from inertial sensors can be used to develop hybrid deep learning (HDL) system to facilitate rehabilitation process [40].”

New reference:

  1. Bijalwan, V.; Semwal, V.B.; Singh, G.; Mandal, T.K. HDL-PSR: Modelling Spatio-Temporal Features Using Hybrid Deep Learning Approach for Post-Stroke Rehabilitation. Neural Process. Lett. 2022, doi:10.1007/s11063-022-10744-6.
